# D4Z4 Hypomethylation in Human Germ Cells

**DOI:** 10.3390/cells13171497

**Published:** 2024-09-06

**Authors:** Ramya Potabattula, Jana Durackova, Sarah Kießling, Alina Michler, Thomas Hahn, Martin Schorsch, Tom Trapphoff, Stefan Dieterle, Thomas Haaf

**Affiliations:** 1Institute of Human Genetics, Julius Maximilians University, 97074 Wuerzburg, Germany; ramya.potabattula@uni-wuerzburg.de (R.P.); jana.durackova@uni-wuerzburg.de (J.D.); sarah.kiessling@stud-mail.uni-wuerzburg.de (S.K.); alina.michler@stud-mail.uni-wuerzburg.de (A.M.); 2Fertility Center Wiesbaden, 65189 Wiesbaden, Germany; th.hahn@mail.de (T.H.); martin.schorsch@gmx.de (M.S.); 3Fertility Center Dortmund, 44135 Dortmund, Germany; trapphoff@kinderwunschzentrum.org (T.T.); dieterle@kinderwunschzentrum.org (S.D.); 4Division of Reproductive Medicine and Infertility, Department of Obstetrics and Gynecology, Witten/Herdecke University, 44135 Dortmund, Germany

**Keywords:** DNA methylation, DUX4, D4Z4, facioscapulohumeral muscular dystrophy, germinal vesicle oocyte, sperm

## Abstract

Expression of the double homeobox 4 (*DUX4*) transcription factor is highly regulated in early embryogenesis and is subsequently epigenetically silenced. Ectopic expression of *DUX4* due to hypomethylation of the D4Z4 repeat array on permissive chromosome 4q35 alleles is associated with facioscapulohumeral muscular dystrophy (FSHD). In peripheral blood samples from 188 healthy individuals, D4Z4 methylation was highly variable, ranging from 19% to 76%, and was not affected by age. In 48 FSHD2 patients, D4Z4 methylation varied from 3% to 30%. Given that *DUX4* is one of the earliest transcribed genes after fertilization, the D4Z4 array is expected to be unmethylated in mature germ cells. Deep bisulfite sequencing of 188 mainly normozoospermic sperm samples revealed an average methylation of 2.5% (range 0.3–22%). Overall, the vast majority (78%) of individual sperm cells displayed no methylation at all. In contrast, only 19 (17.5%) of 109 individual germinal vesicle (GV) oocytes displayed D4Z4 methylation <2.5%. However, it is not unexpected that immature GV oocytes which are not usable for assisted reproduction are endowed with D4Z4 (up to 74%) hypermethylation and/or abnormal (*PEG3* and *GTL2*) imprints. Although not significant, it is interesting to note that the pregnancy rate after assisted reproduction was higher for donors of sperm samples and oocytes with <2.5% methylation.

## 1. Introduction

With a prevalence of about 1 in 20,000, facioscapulohumeral muscular dystrophy (FSHD) is one of the most frequent autosomal dominant muscular dystrophies [1,2]. Symptoms (weakness of facial and shoulder girdle muscles) usually start in the second decade of life. The vast majority of patients suffers from FSHD1, which is caused by contraction of the D4Z4 tandem repeat array on chromosome 4q35.2 from 11 to 150 down to 1–10 repeats. In about 5% of cases, mutations in the structural maintenance of chromosomes hinge-domain-containing protein 1 (*SMCHD1*) gene on chromosome 18q11.32 cause demethylation of all D4Z4 repeats (on chromosomes 4 and 10) and FSHD2 [3,4,5,6,7]. Each D4Z4 repeat contains a copy of the double homeobox protein 4 (*DUX4*) gene, encoding a sequence-specific transcription factor. However, only the last repeat produces a functional *DUX4* transcript, provided it lies on a permissive allele with a unique *DUX4*”ATTAAA” polyadenylation signal outside of the D4Z4 array. Both FSHD1 and FSHD2 require this pathogenic haplotype 4qA, containing the pLAM region with a polyadenylation signal. The non-pathogenic haplotype 4qB and the highly similar (95% sequence identity) D4Z4 arrays on chromosome 10 do not contain a *DUX4* polyadenylation signal and, therefore, are not involved in FSHD pathogenesis. In FSHD1, contraction of the chromosome 4 array leads to a less-condensed local chromatin structure, facilitating the expression of the last *DUX4* copy from permissive alleles. In FSHD2, the haploinsufficiency of *SMCHD1* affects D4Z4 chromatin condensation, leading to the de-repression of *DUX4* transcription. In FSHD1, D4Z4 chromatin relaxation is limited to the contracted allele, whereas in FSHD2, the chromatin relaxation and D4Z4 hypomethylation occur on all chromosome 4 and 10 repeats [3,4,5,6,7]. Ectopic *DUX4* expression is cytotoxic, causing muscle cell death in FSHD patients [8,9,10].

*DUX4* is typically expressed in early human and mouse embryos [11,12]. It acts as a transcriptome and chromatin modifier, priming embryonic genome activation, which occurs in minor and major transcription waves [12,13]. *DUX4* knockdown in human embryos induces changes in embryonic genome activation, but does not terminate development [13]. Similarly, *Dux4^−/−^* knockout mice show defects in pre- and post-implantation development and late embryo mortality, but some *Dux4*^−/−^ pups are viable [14,15]. *DUX4*-binding motifs are enriched in cleavage-stage genes for human genome activation. In addition, *DUX4* activates transcription from non-coding elements, i.e., certain long terminal repeat families in human embryos, which may function as cis-acting regulatory elements during embryonic lineage specification [16,17,18].

*DUX4* transcripts are among the earliest transcribed genes in human and mouse embryos [11,12,17]. They are important for embryonic genome activation, but are not strictly essential for embryogenesis [13,14,15]. Germline reprogramming of the gamete epigenome is generally thought to regulate gene transcription in the early embryo [19,20]. The main aim of our study was to determine the D4Z4 methylation patterns in sperm and oocytes, which may have an effect on early embryogenesis and the chances to establish a pregnancy. Hypomethylation of the D4Z4 array in FSHD2 patients is well known [3,4,5,6,7]; however, in diagnostics, the results of methylation analysis are often difficult to interpret. Therefore, we used the highly accurate bisulfite pyrosequencing (BPS) technique to determine methylation variation and possible confounding factors (age and sex) in blood samples of FSHD2 patients and healthy controls.

## 2. Materials and Methods

### 2.1. Study Samples

The DNA samples from peripheral blood were anonymized excess materials from genetic diagnostics. Informed consent was obtained from all individuals participating in this study. In total, 94 males and 94 females without mutation in predictive diagnostics (for breast cancer and other non-muscular diseases) were considered as healthy controls. In addition, we collected 27 males and 21 females with FSHD2, carrying a pathogenic mutation in the *SMCHD1* gene.

Semen samples were collected at the Fertility Center Wiesbaden. The clinical parameters of the 188 samples used in this study are summarized in Appendix A. After infertility treatment by in vitro fertilization (IVF)/intracytoplasmatic sperm injection (ICSI), the left-over swim-up sperm fraction (excess material) was pseudonymized and snap-frozen at −80 °C until further use. To eliminate contamination by bacteria, lymphocytes, epithelial, and other somatic cells, the swim-up sperm samples were gently thawed and purified further by density gradients PureSperm 80 and 40 (Nidacon, Mölndal, Sweden).

Oocytes were collected at the Fertility Center Dortmund. Following ovarian stimulation and human chorionic gonadotropin priming, immature human germinal vesicle (GV) oocytes were obtained using oocyte retrieval from large antral follicles from women undergoing ICSI treatment due to male factor infertility. Exclusion criteria included women diagnosed with endometriosis, polycystic ovary syndrome, cancer, and those with an anti-Mullerian hormone level <1 ng/mL. Altogether, 109 human GV oocytes were collected between 2021 and 2023 and pseudonymized at the Fertility Center Dortmund. Informed consent was obtained from all 53 women participating in this study. Clinical parameters of the 53 female donors from 56 oocyte pick-ups (OPUs) are summarized in Appendix A. Please note that some of the 53 donors contributed multiple oocytes and/or had more than one OPU. All methods were performed in accordance with the relevant guidelines and regulations. To prevent contamination by somatic cells, the oocytes were separated from the granulosa cells, washed with phosphate-buffered saline, and preserved at −80 °C for further analysis.

### 2.2. DNA Isolation and Conversion

DNA from peripheral blood samples was isolated using the classical salting-out method. To isolate bulk sperm DNA, the purified sperm cells were resuspended in 300 µL buffer composed of 5 mL of 5 M NaCl, 5 mL of 1 M Tris-HCl (pH 8), 5 mL of 10% SDS (pH 7.2), 1 mL of 0.5 M EDTA (pH 8), 1 mL of 100% β-mercaptoethanol, and 33 mL of H_2_O. Additionally, 100 µL of proteinase K (20 mg/mL; 600 mAU/mL; Qiagen, Hilden, Germany) were added. This mixture was then incubated at 56 °C for 2 h. The sperm DNA was extracted using the DNeasy Blood and Tissue kit (Qiagen). The DNA concentration and purity level were determined using a NanoDrop 2000c spectrophotometer (Thermo Scientific, Waltham, MA, USA). The DNA underwent bisulfite conversion with the EpiTect Fast 96 Bisulfite kit (Qiagen), and the converted DNA was preserved at −20 °C for subsequent use. 

DNA from individual human oocytes was isolated and bisulfite-converted using the EZ DNA Methylation Direct Kit (Zymo Research Corporation, Irvine, CA, USA), which is specifically designed and suited for minimal DNA quantities. Briefly, 10 µL of 2× digestion buffer and 1 µL of proteinase K (20 µg/µL) were added to a tube containing a single oocyte. The mixture was then incubated at 50 °C for 20 min, followed by adding 130 µL of bisulfite conversion mix to each sample. The conversion process entailed heating in a thermal cycler at 98 °C for 8 min and then at 64 °C for 3.5 h. Following conversion, the DNA was purified using a spin column and eluted in 10 µL of elution buffer. The bisulfite conversion efficiency is estimated to exceed 99%, with a DNA recovery rate of about 80%. 

### 2.3. Bisulfite Pyrosequencing

Primers for multiplex and singleplex PCR, along with pyrosequencing primers (Appendix A) for the human *DUX4* locus and the germline imprinting control regions of *PEG3* and *GTL2*, were designed using the PyroMark Assay Design 2.0 software (Qiagen). 

The blood samples of healthy individuals and FSHD2 patients were subjected to *DUX4* PCR using ~25 ng of bisulfite-converted DNA, inner forward and reverse primers, and FastStart Taq DNA polymerase (Roche Diagnostics, Mannheim, Germany). The amplification protocol included an initial denaturation step at 95 °C for 5 min, followed by 40 cycles of 95 °C for 30 s, primer-specific annealing temperature of 58 °C for 30 s and 72 °C for 45 s, and a final extension at 72 °C for 10 min. PyroMark Q24 software (Qiagen) was used to perform pyrosequencing on the PyroMark Q24 MDx instrument (Qiagen).

Individual oocytes were first subjected to a multiplex PCR to amplify the *DUX4*, *PEG3*, and *GTL2* loci. The 25 µL reaction mixture included 2.5 µL of 10x PCR buffer with MgCl_2_, 0.5 µL of 10 mM dNTP mixture, 0.2 µL of 5 U/µL FastStart Taq DNA polymerase, 1.25 µL each of 10 pmol/mL forward and reverse outer primers, and 10 µL of bisulfite-converted template DNA. The amplification protocol was initiated with a denaturation phase at 95 °C for 5 min, followed by 35 cycles of 95 °C for 30 s, an annealing temperature of 59 °C for 30 s, an extension at 72 °C for 45 s, with a final extension period at 72 °C for 10 min. Nested singleplex PCRs for each of the three studied amplicons were carried out using 3 µL of the first-round multiplex PCR product as a template. Except for using the inner forward and reverse primers, all other components of the reaction mixture were the same as above. Other than the annealing temperatures, which were set at 58 °C for *DUX4*, 60 °C for *GTL2*, and 57 °C for *PEG3*, the cycling parameters were consistent with those used in the multiplex PCR. Ultimately, pyrosequencing was performed on individual human oocytes utilizing the PyroMark Q24 MDx system.

The BPS of genomic DNA samples (from blood) allows for the accurate quantification of mean methylation of 9 contiguous CpGs in the D4Z4 target region. In our experience, with various amplicons, the methylation difference between technical replicates (including bisulfite conversion) is in the order of 1–3 percentage points. For the BPS of single oocytes, we cannot do technical replicates. For some donors, the D4Z4 methylation measurements of multiple oocytes differ only by a few percent (Appendix A). This may correspond to technical variation between replicates.

### 2.4. Deep Bisulfite Sequencing (DBS)

DBS was carried out on human sperm samples using *DUX4* primers (Appendix A), covering all 30 CpGs in the DR1 region [4]. The initial PCR reactions were conducted in volumes of 50 µL, comprising 5 µL of 10× PCR buffer with MgCl_2_, 1 µL of 10 mM PCR grade nucleotide mix, 0.4 µL of 5 U/µL FastStart Taq DNA polymerase, 2.5 µL each of 10 pmol/mL forward and reverse primers, 2 µL of ~50 ng bisulfite-converted genomic DNA, and 36.6 µL of ddH_2_O. Fully methylated and unmethylated DNA standards (Qiagen) served as controls for assessing the reliability of methylation measurements. The first-round PCR products were cleaned with Agencourt AMPure XP beads (Beckmann Coulter, Krefeld, Germany), quantified using the Qubit dsDNA BR Assay system kit (Invitrogen, Karlsruhe, Germany), and diluted to a concentration of 0.2 ng/µL. In the final PCR, the samples were barcoded using multiple identifiers (MIDs), specifically NEBNext Multiplex Oligos for Illumina (Dual Index Primers Set 1 and 2). The purified and quantified PCR pools were diluted to a concentration of 4 nM, and 3 µL of this dilution from each of the MIDs were pooled together into one final pool for next-generation sequencing (NGS). 

NGS was performed using the MiSeq platform (Illumina, San Diego, CA, USA) and Reagent Kit V2 cartridge (500 cycles; Illumina) according to the manufacturer’s recommendations. The sequencing process was executed using 250 bp paired-end sequencing. Following the run, the reads were processed using an Illumina Genome Analyzer. FASTQ files were further analyzed using Amplikyzer2 software [21], which provides a detailed nucleotide-level analysis and CpG methylation rates at both single nucleotide and regional levels. Only reads with an overall bisulfite conversion rate of >95% were considered further, and downstream processing of Amplikyzer output files and subsequent analyses of methylation rates were performed.

Our DBS assay analyzes 30 contiguous CpGs in the D4Z4 array. The difference in mean methylation of all CpGs between technical replicates is in the order of 1–3 percentage points.

### 2.5. Statistical Analysis

The statistical analysis, encompassing descriptive and inferential aspects, was executed with IBM SPSS software (version 28).

## 3. Results

### 3.1. D4Z4 Methylation in Blood Is Highly Variable

Previously, we established a BPS assay for FSHD2 diagnostics, targeting nine CpGs in the DR1 domain of D4Z4 [6]. Using this assay, we quantified D4Z4 methylation levels in peripheral blood samples of 94 male and 94 female healthy controls with an age range from 1 to 70 years. Methylation was slightly higher (Mann–Whitney U test; *p* = 0.017) in males (Figure 1, blue dots) than in females (Figure 1, red dots). However, overall methylation levels were highly variable and largely overlapping between male (mean ± SD 53 ± 11%; range 23–74%) and female (49 ± 12%; range 19–76%) controls. There was no significant correlation between methylation and age in males (Spearman rho = −0.11; *p* = 0.28) and females (rho = 0.13; *p* = 0.20).

In addition, we analyzed 27 male (Figure 1, green dots) and 21 female (Figure 1, yellow dots) FSHD2 patients with pathogenic *SMCHD1* mutations. There was no significant (Mann–Whitney U test; *p* = 0.51) difference in D4Z4 methylation between male (mean ± SD 10.8 ± 7.5%; range 3–30%) and female (10.8 ± 5.5%; range 4–27%) patients. Similar to controls, methylation was not correlated with age (rho = 0.09; *p* = 0.68 in males and rho = 0.05; *p* = 0.82 in females). As expected, FSHD2 patients showed a highly significant (Mann–Whitney U test; *p* < 0.0001) D4Z4 hypomethylation (11 ± 7%; range 3–30%) compared to controls (51 ± 12%; range 19–76%). From a diagnostic point of view, it is important to emphasize that there is a gray area between FSHD2 and controls. In total, 5 (3%) out of 188 controls displayed methylation values below 25%, and 2 (4%) out of 48 FSHD2 patients displayed values above 25%.

### 3.2. D4Z4 Hypomethylation in Sperm

Since *DUX4* is expressed in human cleavage-stage embryos influencing embryonic genome activation, it is plausible to assume that the D4Z4 array is hypomethylated in sperm. For analyzing D4Z4 methylation at the single allele/sperm level, we developed a DBS assay for the complete DR1 region containing 30 CpGs in 259 bp. Using this assay, we analyzed 188 sperm samples. Although all samples were from men attending a fertility center, the semen parameters were largely normal (Appendix A). In total, 112 (60%) samples were used for IVF and 76 (40%) for ICSI. In total, 103 (55%) samples led to a pregnancy by IVF/ICSI treatment, whereas 85 (45%) did not. With an average methylation of 2.5 ± 3.2% (range 0.3–22%) across all 188 samples (representing almost 11 million individual reads), the sperm D4Z4 array was strongly hypomethylated, compared to somatic tissue (blood). The vast majority (78.1%) of reads (each representing a single sperm) displayed no methylation at all. In 11.1% of reads, one of 30 analyzed CpGs was methylated. In total, 6.5% of reads displayed 2–5, 2.6% displayed 6–10, and 1.6% displayed more than 10 methylated CpGs.

The scatter plot in Figure 2 presents the mean methylation values of the DR1 region in 188 sperm samples. There is no correlation between methylation and donor age (Spearman rho = 0.11; *p* = 0.12). To test whether D4Z4 methylation has an impact on assisted reproduction outcome, we more-or-less arbitrarily defined a threshold of <2.5% for sperm samples with normal hypomethylation patterns. This means that, on average, the D4Z4 array in an individual sperm of this donor contains no or only a single methylated CpG (of 30 analyzed CpGs). We then compared the pregnancy rates between samples with <2.5% and ≥2.5% methylation, respectively. Using IVF/ICSI, 76 (56.3%) out of 133 samples with <2.5% methylation and 27 (50.9%) out of 53 samples with ≥2.5% resulted in a clinical pregnancy (Figure 3). This between-group difference in pregnancy rates is not significant (χ^2^ test; *p* = 0.51). However, in this context, it is important to mention that each of the 188 analyzed samples contains millions of individual sperm. Even the five sperm samples with the highest mean methylation values (ranging from 11% to 22%) were endowed with 27% to 41% of completely unmethylated sperm alleles (Figure 4).

### 3.3. D4Z4 Methylation in Oocytes

We established a multiplex PCR assay including the oppositely imprinted control regions of *PEG3* (maternally methylated) and *GTL2* (paternally methylated), as well as nine CpGs in the D4Z4 array. BPS was used to study methylation of the three target regions in immature GV oocytes, each containing four alleles (2n4c). D4Z4 methylation could be quantified in 109 individual oocytes. Methylation was highly variable (mean ± SD 16.9 ± 17.1%), ranging from 0.5% to 74% (Figure 5). There was no correlation between oocyte D4Z4 methylation and the donor’s age (Spearman rho = −0.07; *p* = 0.39). In 91 and 92 of 109 oocytes, respectively, we also obtained *PEG3* and *GTL2* methylation values. As expected for a paternally and a maternally imprinted gene, *GTL2* was hypomethylated (12.3 ± 21.6%) and *PEG3* was hypermethylated (77.8 ± 32.5%) in oocytes. A considerable number (30 of 109; 27.5%) of oocytes displayed an abnormal methylation imprint(s). Imprinting errors of either *PEG3* (<80% methylation) or *GTL2* (>20% methylation) and D4Z4 hypermethylation (≥2.5%) appeared to occur independently of each other. Thirty-four women contributed multiple oocytes. Although different oocytes from the same woman and OPU usually displayed similar D4Z4 methylation values, some donors had oocytes differing by 20–50% in D4Z4 methylation (Appendix A).

For the sake of simplicity, we also used a 2.5% threshold for D4Z4 methylation to classify oocytes in two groups, 19 oocytes with low (<2.5%) and 90 oocytes with increased (≥2.5%) D4Z4 methylation. In group 1 with strong D4Z4 hypomethylation, 3 (16%) of 19 oocytes displayed imprinting errors, in group 2 with increased D4Z4 methylation, 27 (30%) of 90 oocytes. Although the 2.5% threshold is arbitrarily defined, our results suggest that relatively few (14 of 109; 13%) GV oocytes have normal methylation patterns for all three analyzed regions. 

Since some donors contributed multiple oocytes per OPU, patients were finally classified into two groups to assess the clinical pregnancy rate per OPU. Patients with at least one GV oocyte with <2.5% D4Z4 methylation per OPU were assigned to group 1 and patients with no GV oocyte with <2.5% D4Z4 methylation per OPU were assigned to group 2. The clinical pregnancy rate per OPU of donors of oocytes with <2.5% D4Z4 methylation was higher (6 of 17 OPU; 35%) than that of donors of oocytes with ≥2.5% methylation (8 of 39 OPU; 20%) (Figure 6 and Appendix A). Although this observation supports the hypothesis that oocytes with <2.5% methylation have a higher developmental potential, due to the small sample, there is no significant (χ^2^ test; *p* = 0.24) between-group difference.

## 4. Discussion

### 4.1. D4Z4 Methylation and FSHD

Expression of the transcription factor *DUX4* is restricted to early embryogenesis, and the *DUX4* binding motifs are enriched in genes for embryonic genome activation, pre- and post-implantation development [13,17]. With the notable exception of the testes and thymus, *DUX4* expression is epigenetically silenced by DNA methylation and histone modification (H3K9me3), packing the D4Z4 array into constitutive heterochromatin in adult tissues [22]. Aberrant reactivation of *DUX4* in somatic tissues has been associated with different diseases, including virus infection, various neoplasms, and, most importantly, FSHD [23]. Ectopic *DUX4* expression activates pathways for oxidative stress, DNA damage, inflammation, and apoptosis in skeletal muscle cells [8,9,10]. In FSHD1, a contraction of the D4Z4 repeat on chromosome 4q35 makes 4qA alleles with a *DUX4* polyadenylation signal permissive for *DUX4* expression in skeletal muscles. In patients without D4Z4 contractions, the disease is caused by mutations in *SMCHD1* and rarely other genes (*LRIF1* and *DNMT3B*) involved in the epigenetic silencing of the FSHD locus [3,4,5,6,7].

Consistent with the epigenetic silencing of the FSHD locus, the D4Z4 array (on both chromosome 4 and 10) is hypermethylated in blood. Although hypomethylation of the D4Z4 array in FSHD2 patients is well known [3,4,5,6,7] and frequently used for diagnostics, we determined the methylation variation in blood of normal healthy controls and FSHD2 patients with pathogenic *SMCHD1* mutations. BPS can quantify average D4Z4 methylation with an accuracy of 1–3 percentage points (variation between technical replicates). There were no detectable age effects on D4Z4 methylation, and methylation variation was highly similar between males and females. Therefore, we can exclude age and sex as confounding factors in FSHD diagnostics. Although FSHD2 samples were significantly hypomethylated (11 ± 7%; range 3–30%) compared to controls (51 ± 12%; range 19–76%), the 25% threshold, which is frequently used in diagnostics, does not clearly separate FSHD2 patients and controls. Samples with 20–30% methylation have to be interpreted with caution. 

The enormous methylation variation in both FSHD2 patients and controls may be explained by the fact that BPS and other diagnostic methods (i.e., NGS) measure the average methylation of all repeats on chromosomes 4 and 10. The number of repeats in normal individuals is highly variable, ranging from 10 to 150 on chromosome 4 and similarly on chromosome 10 [3,4,5,6,7]. Chromosome 4B and chromosome 10 alleles, which do not contain a *DUX4* polyadenylation signal, contribute to average D4Z4 methylation, but not to disease pathogenesis. In FSHD1, contraction of the 4qA allele is associated with hypomethylation of the last repeat in the array, whereas in FSHD2, all chromosome 4 and 10 alleles are hypomethylated [4,6]. FSHD1 and FSHD2 appear to form a disease continuum. In general, FSHD1 patients exhibit 1–10 D4Z4 repeat and FSHD2 patients 11–20. Patients carrying 9–10 repeats can have the genetic and epigenetic characteristics of both FSHD1 and 2 [24].

### 4.2. D4Z4 Methylation in Sperm

Its expression in early cleavage-stage embryos [11,12] suggests a role for *DUX4* in gene regulation after fertilization. D4Z4 arrays are hypomethylated in sperm, adopting an open chromatin structure which allows for *DUX4* transcription in early embryos. Consistent with our working hypothesis, the mean D4Z4 methylation in 188 sperm samples measured using DBS was 2.5 ± 3.2% (range 0.3–22%). This is a very strong hypomethylation, even compared to the hypomethylated D4Z4 array (11 ± 7%; range 3–30%) in somatic blood tissue of FSHD2 patients. In almost 80% of analyzed sperm alleles (of 188 samples), all 30 CpGs in the DR1 region were completely unmethylated. Even extreme outliers with >10% mean D4Z4 methylation displayed large numbers (27–41%) of completely unmethylated alleles. Collectively, these results suggest that “normal” sperm display very low mean D4Z4 methylation, which may be functionally important for *DUX4* expression in early embryos. Accumulating evidence suggests that germ-cell methylation influences the transcriptional activation of genes in embryo development [19,20,25,26]. Although the 2.5% threshold for normal D4Z4 methylation is somewhat arbitrary, it was chosen because, in bulk sperm samples, mean D4Z4 methylation is 2.5%. This implies that, at the individual sperm level, at most, 1 of 30 CpGs is methylated (single CpG methylation error). Using IVF/ICSI, the pregnancy rate of sperm samples with <2.5% mean methylation is somewhat higher (56%) than for samples with ≥2.5% (51%). Considering that, in most sperm samples, the vast majority of alleles are completely unmethylated, it is not unexpected that the effect of D4Z4 methylation on IVF/ICSI outcome is not significant. In the end, it takes only a single sperm to fertilize an egg.

### 4.3. D4Z4 Methylation in Oocytes

Compared to DBS on bulk sperm samples, analysis of single oocytes is more challenging. Using multiplex PCR and BPS, we were able to quantify D4Z4 methylation in 109 individual oocytes. For most oocytes, we also obtained methylation of the paternally imprinted *GTL2* and the maternally imprinted *PEG3* gene. Since, due to ethical reasons, we only used immature GV oocytes which were not suitable for ICSI, a relatively large number (30 of 109; 28%) of oocytes showed imprinting defects, which are due to errors in imprint establishment/maintenance during oocyte development and are associated with poor oocyte quality [27]. Only 19 (17.4%) of 109 GV oocytes displayed low D4Z4 methylation, ranging from 0.5% to 2.5%, whereas 90 (82.6%) GV oocytes had methylation values from ≥2.5% to 74%. This may largely be due to the compromised quality of our study material. Nevertheless, we propose that, similar to male germ cells, “normal” mature metaphase II oocytes should have low (<2.5%) D4Z4 methylation, enabling *DUX4* expression in early cleavage-stage embryos. Interestingly, in our study, neither D4Z4 methylation nor abnormal imprinting correlated with maternal age. Instead, the observed methylation defects may be due to ovulation induction and other stressors during oocyte development [27]. There was a trend for a higher clinical pregnancy rate per OPU after ICSI treatment in donors of oocytes with low (<2.5%) D4Z4 methylation than in donors of oocytes with ≥2.5% methylation. However, in this context, it is important that oocytes from the same donors may differ in D4Z4 methylation and developmental potential, especially between mature metaphase II oocytes used for ICSI treatment and immature GV oocytes analyzed here.

### 4.4. Limitations

We show that D4Z4 is strongly hypomethylated in sperm and oocytes, but there is only circumstantial evidence that this D4Z4 hypomethylation is important for the function of *DUX4* in early embryos. In different species (from frogs to humans), it has been shown that germ cell epigenomes regulate transcription of a set of developmentally important genes in early embryos [19,20,25,26]. In general, hypomethylation in gametes is associated with transcriptional activation of genes in early embryo development. In FSHD2 patients, it has been demonstrated that D4Z4 hypomethylation in blood is associated with chromatin relaxation and ectopic *DUX4* expression.

Due to ethical problems, we were not able to collect a larger number of GV oocytes or even mature oocytes. We conclude that the pregnancy rate after assisted reproduction is higher for donors of oocytes with unmethylated D4Z4 arrays; however, these results are not significant. This may not only be due to small sample size. If we determine the D4Z4 methylation status in a given sperm cell or oocyte, this cell cannot be used for fertilization. Each individual bulk sperm sample contains millions of sperm with unmethylated D4Z4, along with a variable percentage of sperm with methylated D4Z4 arrays. Similarly, different oocytes of the same donor can have both hypo- and hypermethylated D4Z4 arrays. We cannot have information on the D4Z4 methylation status in both germ cells and the resulting embryo.

## 5. Conclusions

Mean methylation of chromosome 4- and 10-derived D4Z4 repeats is highly variable (ranging from 19% to 76%) in somatic tissue (blood) of healthy individuals, independent of sex and age. Although FISHD2 patients with pathogenic *SMCHD1* mutation show significant D4Z4 hypomethylation, FSHD2 and controls overlap in the 20–30% methylation range. In contrast, D4Z4 is strongly hypomethylated (<2.5%) in the majority of sperm and a proportion of immature GV oocytes. We propose that D4Z4 hypomethylation, and by extrapolation, a relaxed chromatin structure in germ cells, facilitates *DUX4* transcription and embryonic genome activation after fertilization.

## Figures and Tables

**Figure 1 cells-13-01497-f001:**
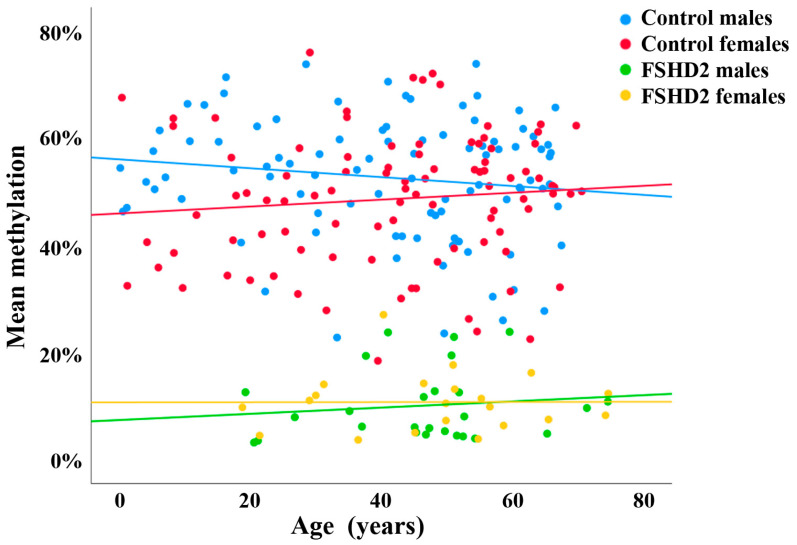
Methylation variation of D4Z4 in healthy individuals and FSHD2 patients, determined using PBS. Scatter plot showing blood D4Z4 methylation levels in 94 male (blue dots) and 94 female (red dots) healthy controls, as well as in 27 male (green dots) and 21 female (yellow dots) FSHD2 patients. There is no age effect in controls and FSHD patients, respectively. However, there is significant hypomethylation in FSHD2 compared to controls.

**Figure 2 cells-13-01497-f002:**
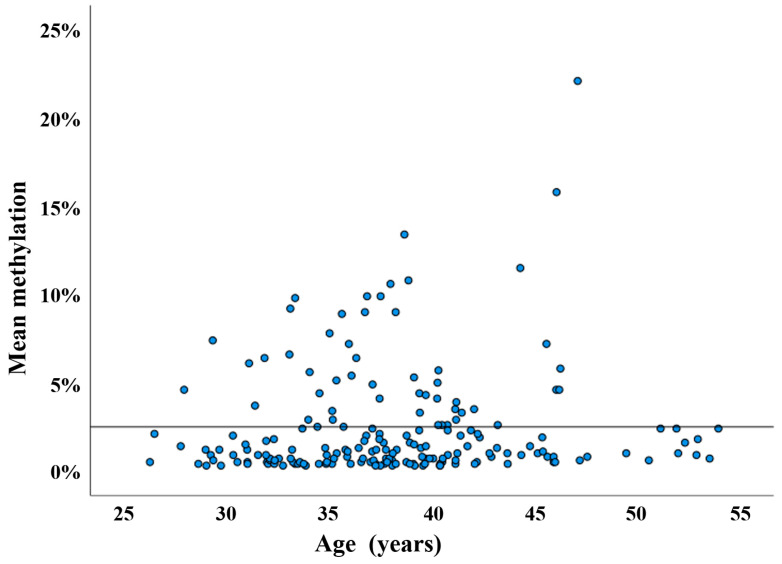
D4Z4 methylation in human sperm samples, determined using DBS. Scatter plot showing D4Z4 methylation levels in 188 sperm samples (blue dots) from males with an age range from 26 to 54 years. The horizontal line indicates the 2.5% methylation threshold. The vast majority (135 of 188) of samples have methylation values between 0% and 2.5%.

**Figure 3 cells-13-01497-f003:**
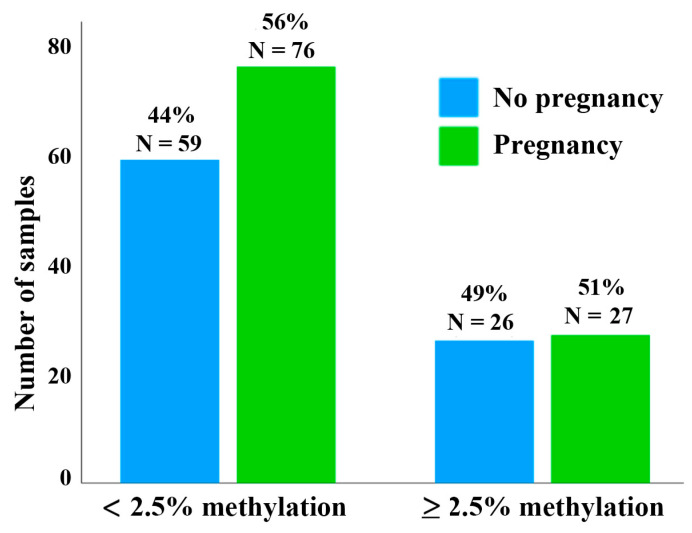
Pregnancy rate and sperm D4Z4 methylation. The bar diagrams show the IVF/ICSI pregnancy rate using 135 sperm samples with <2.5% D4Z4 methylation compared to 53 samples with ≥2.5% methylation.

**Figure 4 cells-13-01497-f004:**
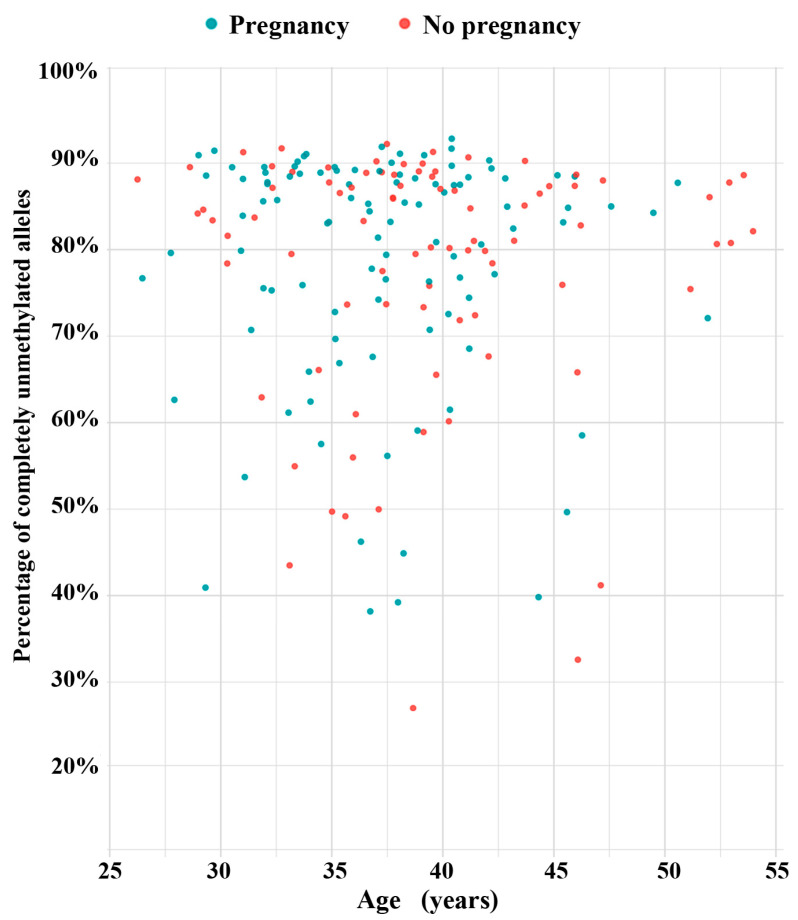
Percentage of completely unmethylated alleles in 188 sperm samples. Blue dots indicate 103 samples resulting in an IVF/ICSI pregnancy, red dots indicate 85 samples without pregnancy. The five sperm samples with the highest mean methylation values (13.5%, 16%, 11%, 11.5%, and 22%) were endowed with 27%, 33%, 39%, 40%, and 41% of completely unmethylated alleles.

**Figure 5 cells-13-01497-f005:**
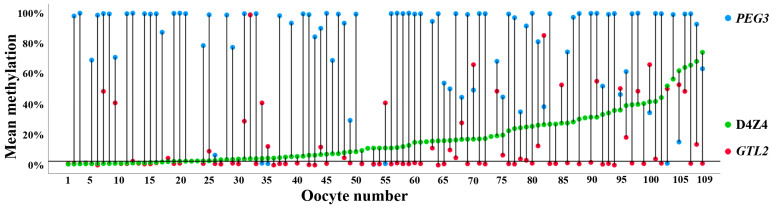
D4Z4, *PEG3*, and *GTL2* methylation in 109 GV oocytes, determined using multiplex PCR and BPS. Each vertical line represents an individual oocyte numbered on the x axis from 1 to 109 with increasing D4Z4 methylation values. D4Z4 methylation is indicated by green dots, *PEG3* methylation by blue dots, and *GTL2* methylation by red dots. The horizontal line indicates the 2.5% methylation threshold. A considerable number of oocytes show abnormal methylation imprints, i.e., in oocyte no. 32, *GTL2* is >95% hypermethylated, and in oocytes 34 and 35, *PEG3* is <2.5% hypomethylated. Please note that in contrast to the D4Z4 repeat, we did not obtain amplification products for the methylation analysis of the single-copy genes, *PEG3* and *GTL2*, in all oocytes.

**Figure 6 cells-13-01497-f006:**
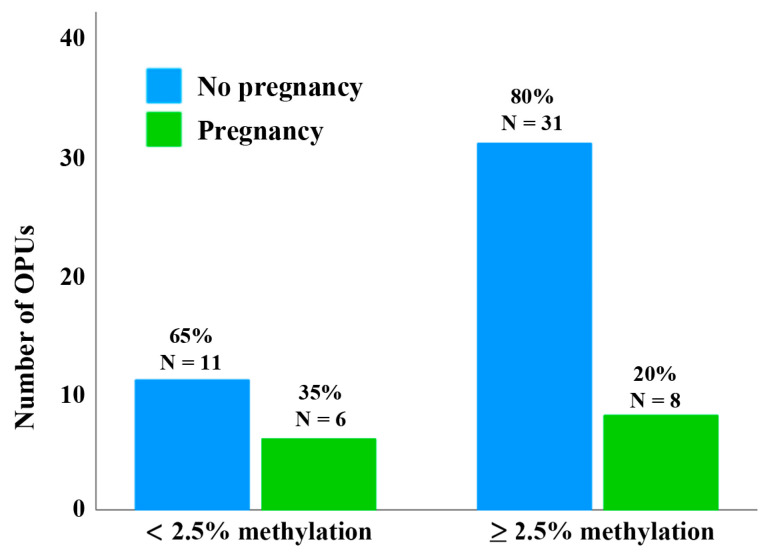
Clinical pregnancy rates per OPU of donors of oocytes with <2.5% and ≥2.5% D4Z4 methylation. From the overall 56 OPUs, 17 OPUs had at least one GV oocyte with <2.5% D4Z4 methylation, while in 39 OPUs, no GV oocyte with <2.5% D4Z4 methylation was present. The clinical pregnancy rate per OPU was higher in donors of lowly methylated oocytes (6 of 17 OPU; 35%) compared to the group with ≥2.5% methylation (8 of 39 OPU; 20%). Please note that some of the 53 donors had more than one OPU.

## Data Availability

All data are presented in the article and its online Appendix A.

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
