# Peer review of "D4Z4 Hypomethylation in Human Germ Cells"

_cells, 2024, doi:10.3390/cells13171497_

Round 1

Reviewer 1 Report

Comments and Suggestions for Authors

This manuscript submitted by Potabattula et al. describes D4Z4 hypomethylation in human germ cells. D4Z4 tandem repeat shrinkage typically causes facioscapulohumeral muscular dystrophy (FSHD), and about 5% of cases result in hypomethylation of all D4Z4 repeats in the structural maintenance of chromosomes hinge-domain-containing protein (SMCHD1) gene. Dux4 is the only functional transcript with a polyadenylation signal. In this report, the authors analyzed the hypomethylation pattern of the D4Z4 array in gametes and its expression after fertilization in both gametes and somatic tissue.

Most of the results are not statistically significant, as the authors do not find a correlation between D4Z4 methylation levels in blood, sperm, and oocytes. Specifically, there are no significant differences in the methylation of D4Z4 between males and females across different age groups. Similar results were observed for male vs. female FSHD2 patients, whose D4Z4 is hypomethylated and statistically different from that of healthy patients. Additionally, D4Z4 methylation in human samples is highly variable across ages. Most importantly, there is no correlation between D4Z4 methylation levels and pregnancy outcomes, and they do not play a significant role in this context.

Overall, this report attempts to identify the methylation pattern in D4Z4 arrays in human germ cells, but the results are highly variable. A major concern about this paper is the novelty of the findings and the sample size, which could be addressed in the conclusion or discussion section.

Major Concerns:

  1. Most of the results are not statistically significant. Hence, it is unclear whether the D4Z4 methylation pattern plays a critical role. The authors have not addressed whether the function of D4Z4 is attributed to this methylation pattern or if it carries any other functions, such as through its binding domains and partners.
  2. The sample size: The sample size used in this paper seems quite small, making it difficult to draw conclusive results. Additionally, the sample size varies across different sections of the study.

Minor Comments:

  1. In the methods section or figure legends, the authors should include information on whether any technical replicates were carried out.
  2. In Figure 5, not all methylation values are reported for some oocytes. For example, between oocyte numbers 20-25, D4Z4 methylation values are reported, but not for GTL2 or PEG3.
Comments on the Quality of English Language

Minor editing is required

Author Response

This manuscript submitted by Potabattula et al. describes D4Z4 hypomethylation in human germ cells. D4Z4 tandem repeat shrinkage typically causes facioscapulohumeral muscular dystrophy (FSHD), and about 5% of cases result in hypomethylation of all D4Z4 repeats in the structural maintenance of chromosomes hinge-domain-containing protein (SMCHD1) gene. Dux4 is the only functional transcript with a polyadenylation signal. In this report, the authors analyzed the hypomethylation pattern of the D4Z4 array in gametes and its expression after fertilization in both gametes and somatic tissue.

Most of the results are not statistically significant, as the authors do not find a correlation between D4Z4 methylation levels in blood, sperm, and oocytes. Specifically, there are no significant differences in the methylation of D4Z4 between males and females across different age groups. Similar results were observed for male vs. female FSHD2 patients, whose D4Z4 is hypomethylated and statistically different from that of healthy patients. Additionally, D4Z4 methylation in human samples is highly variable across ages. Most importantly, there is no correlation between D4Z4 methylation levels and pregnancy outcomes, and they do not play a significant role in this context.

Overall, this report attempts to identify the methylation pattern in D4Z4 arrays in human germ cells, but the results are highly variable. A major concern about this paper is the novelty of the findings and the sample size, which could be addressed in the conclusion or discussion section.

RESPONSE:   Our bisulfite (pyro)sequencing assays are quite accurate (1-3% difference between technical replicates), excluding that the observed methylation variation is a technical artefact. The hypomethylation od the D4Z4 array in FSHD2 patients is well known and used for the diagnostics of this disease. Our main intention to analyze D4Z4 blood methylation was to show the enormous D4Z4 methylation variation in both controls and FSHD2 patients and that there is an overlap between both groups. Moreover, for diagnostics it is helpful to know that D4Z4 methylation is not affected by sex or age. Similar to somatic tissues, there is no age effect in gametes. This implies that “abnormal” methylation pattern in gametes are due to other (stochastic and environmental) factors, i.e. superovulation. Although not significant, our results promote the idea that the pregnancy rate after ART is higher for donors of germ cells with unmethylated D4Z4 array. The Limitations of our study are discussed.

Major Concerns:

1. Most of the results are not statistically significant. Hence, it is unclear whether the D4Z4 methylation pattern plays a critical role. The authors have not addressed whether the function of D4Z4 is attributed to this methylation pattern or if it carries any other functions, such as through its binding domains and partners.

RESPONSE:   We did not and according to the German embryo protection law cannot study DUX4 in early embryos. We have stated this clearly in the Limitations section. However, there is evidence from the literature that D4Z4 hypomethylation is linked to DUX4 expression.  (1) D4Z4 hypomethylation in FSHD2 is associated with chromatin relaxation and ectopic expression of DUX4.  (2) In different species (from frogs to humans) it has been shown that germ-cell epigenomes regulate transcription of a set of developmentally important genes in early embryos (Jenkins & Carrell, 2012; Smith et al., 2012; Teperek et al., 2016; Dittrich et al., 2024). In general, hypomethylation of a given gene in gametes is associated with transcriptional activation in early embryo development. Considering that DUX4 is among the earliest transcribed gene and its role in embryonic genome activation, we propose that D4Z4 methylation in germ cells helps DUX4 to play its appropriate role in early embryogenesis.

2. The sample size: The sample size used in this paper seems quite small, making it difficult to draw conclusive results. Additionally, the sample size varies across different sections of the study.

RESPONSE:   For D4Z4 methylation in blood samples, we analyzed 94 male and 94 female healthy individuals as well as 48 patients with FSHD2, using bisulfite pyrosequencing. This is a reasonable sample size for this kind of studies, in particular for a rare disease like FSHD2. Similarly, the 188 sperm samples analyzed by deep bisulfite sequencing are not a small sample size. The main problem with sample size pertains to the 109 oocytes used for this study. Due to ethical problems, we were not able to collect more GV oocytes or even mature oocytes. Our results promote the idea that the pregnancy rate after assisted reproduction is higher for donors of germ cells with unmethylated D4Z4 array. That we did not achieve significance may not only be due to small sample size. If we determine the D4Z4 methylation status in a sperm cell or oocyte, this cell cannot be used for fertilization. Each individual sperm samples contains millions of sperm with unmethylated D4Z4, along with a variable percentage of sperm with methylated D4Z4 arrays. Similarly, different oocytes of the same donor can have both hypo- and hypomethylated D4Z4 arrays. We cannot have information on the D4Z4 methylation status in both germ cells and the resulting embryo. This was stated clearly in the Limitations.

Minor Comments:

1. In the methods section or figure legends, the authors should include information on whether any technical replicates were carried out.

RESPONSE:   Both bisulfite pyrosequencing (BPS) and deep bisulfite sequencing (DBS) of genomic DNA samples (from blood and sperm) allow accurate quantification of mean methylation of several contiguous CpGs (9 for the BPS and 30 for the DBS assay) in the D4Z4 target region. In our experience with various amplicons, the methylation difference between technical replicates (including bisulfite conversion) is in the order of 1-3 percentage points. For BPS of single oocytes we cannot do technical replicates. For some donors, the D4Z4 methylation measurements of multiple oocytes differ only by a few percent (Figure S1). This may correspond to technical variation between replicates. We have stated this clearly in the Methods section.

2. In Figure 5, not all methylation values are reported for some oocytes. For example, between oocyte numbers 20-25, D4Z4 methylation values are reported, but not for GTL2 or PEG3.

RESPONSE:   Determining the methylation level of the single copy genes GTL2 and PEG3 in individual oocytes is much more challenging than methylation analysis of the D4Z4 repeat array. We did not get amplification products for GTL2 and PEG3 in all oocytes. We have stated this clearly in the figure legend.

Reviewer 2 Report

Comments and Suggestions for Authors

FSHD is one of the most frequent autosomal dominant muscular dystrophies. And it’s already associated with DUX4 and D4Z4. It is potentially interesting that a higher methylation level of D4Z4 in oocytes is associated with an 80% rate of non-pregnancy. The conclusion and significance of this study are not clearly presented.

Major concerns:

1.     The scientific questions that the author intends to address and the significance of the results are not clearly introduced in the Abstract and Introduction sections.

2.     For 3.1 section, since a previous paper (PMID: 23143600) already showed the correlation between D4Z4 hypomethylation in peripheral blood and FSHD1/2, the novel findings of this section are unclear.

3.     For 3.2 and 3.3, this study examines the methylation levels of D4Z4 in oocytes and sperm and attempts to associate these levels with pregnancy rates. It would be beneficial to show the dynamics of D4Z4 methylation levels during embryo development, given the availability of public DNA methylome datasets. In Figure 6, although a higher methylation level of D4Z4 in oocytes is associated with non-pregnancy (80%), it remains unclear whether D4Z4 hypermethylation directly leads to non-pregnancy. Functional verification is necessary to establish causality.

Minor concerns:

There are numerous inappropriate citations in this manuscript, including instances where a large paragraph is overloaded with citations or where non-fundamental sources are cited. For example:

1.     Line 36-41, missing citation of “(FSHD) is one of the most frequent autosomal dominant muscular dystrophies” and “FSHD1 is caused by D4Z4”.

2.     Line 61-62, missing citation of “DUX knockdown in human embryo”.

3.     Line 62-63, “Phenotypes of DUX knockout in mouse embryos” should cite the fundamental works: PMID: 31133747 and PMID: 31591446.

4.     Line 69-70, missing citations.

Author Response

FSHD is one of the most frequent autosomal dominant muscular dystrophies. And it’s already associated with DUX4 and D4Z4. It is potentially interesting that a higher methylation level of D4Z4 in oocytes is associated with an 80% rate of non-pregnancy. The conclusion and significance of this study are not clearly presented.

Major concerns:

1. The scientific questions that the author intends to address and the significance of the results are not clearly introduced in the Abstract and Introduction sections.

RESPONSE:   Due to the 200 words limit only the results can be presented in the abstract. However, we have added the following paragraph at the end of the introduction, defining our research questions.

DUX4 transcripts are among the earliest transcribed genes in human and mouse embryos. They are important for embryonic genome activation but not strictly essential for embryogenesis. Germline reprogramming of the gamete epigenome is generally thought to regulate gene transcription in the early embryo. The main aim of our study was to determine the D4Z4 methylation patterns in sperm and oocytes, which may have an effect on early embryogenesis and the chances to establish a pregnancy. Hypomethylation od the D4Z4 array in FSHD2 patients is well known, however the results of methylation analysis are often difficult to interpret in diagnostics. Therefore, we used the highly accurate bisulfite pyrosequencing technique to determine methylation variation and possible confounding factors (age and sex) in blood samples of FSHD2 patients and healthy controls

2. For 3.1 section, since a previous paper (PMID: 23143600) already showed the correlation between D4Z4 hypomethylation in peripheral blood and FSHD1/2, the novel findings of this section are unclear.

RESPONSE:   The hypomethylation od the D4Z4 array in FSHD2 patients is well known and used for diagnostics of this disease. Our main intention to analyze D4Z4 blood methylation was to show the enormous D4Z4 methylation variation in both controls and FSHD2 patients and that there is an overlap between both groups. Moreover, for diagnostics it is helpful to know that D4Z4 methylation is not affected by sex or age.

3. For 3.2 and 3.3, this study examines the methylation levels of D4Z4 in oocytes and sperm and attempts to associate these levels with pregnancy rates. It would be beneficial to show the dynamics of D4Z4 methylation levels during embryo development, given the availability of public DNA methylome datasets. In Figure 6, although a higher methylation level of D4Z4 in oocytes is associated with non-pregnancy (80%), it remains unclear whether D4Z4 hypermethylation directly leads to non-pregnancy. Functional verification is necessary to establish causality.

RESPONSE:   From our germ-cell analysis we conclude that pregnancy rate after assisted reproduction is higher for donors of germ cells with unmethylated D4Z4 array, however these results are not significant. In our opinion, it is difficult or even impossible to provide additional functional evidence for our hypothesis. If we determine the D4Z4 methylation status in a given sperm cell or oocyte, this cell cannot be used for fertilization. Each individual bulk sperm samples contains millions of sperm with unmethylated D4Z4, along with a variable percentage of sperm with methylated D4Z4 arrays. Similarly, different oocytes of the same donor can have both hypo- and hypomethylated D4Z4 arrays. We cannot have information on the D4Z4 methylation status in both germ cells and the resulting embryo. This was stated clearly in the Limitations.

However, it has been shown in different species (from frogs to humans) that germ-cell epigenomes regulate transcription of a set of developmentally important genes in early embryos. In general, hypomethylation in gametes is associated with transcriptional activation of genes in early embryo development. 

Minor concerns:

There are numerous inappropriate citations in this manuscript, including instances where a large paragraph is overloaded with citations or where non-fundamental sources are cited. For example:

1. Line 36-41, missing citation of “(FSHD) is one of the most frequent autosomal dominant muscular dystrophies” and “FSHD1 is caused by D4Z4”.

RESPONSE:   We have cited Emery, 1991 and the Orphanet, 2024.

2. Line 61-62, missing citation of “DUX knockdown in human embryo”.

RESPONSE:   We have cited  Vuoristo et al., 2022.

3. Line 62-63, “Phenotypes of DUX knockout in mouse embryos” should cite the fundamental works: PMID: 31133747 and PMID: 31591446.

RESPONSE:   We have cited  Chen and Zhang, 2019  and  Guo et al., 2019.

4. Line 69-70, missing citations.

RESPONSE:   We have cited  De Iaco et al., 2017;  Hendrickson et al., 2017;  and  Whiddon et al., 2017.

Round 2

Reviewer 1 Report

Comments and Suggestions for Authors

The authors addressed all the raised concerns